# A Partial Least Squares Structural Equation Modeling of Robotics Implementation for Sustainable Building Projects: A Case in Nigeria

Ahmed Farouk Kineber [1,2,*], Ayodeji Emmanuel Oke [3,4,5], Mohammed Magdy Hamed [6],
Ehab Farouk Rached [7], Ali Elmansoury [7] and Ashraf Alyanbaawi [8]

1   Department of Civil Engineering, College of Engineering in Al-Kharj, Prince Sattam Bin Abdulaziz University, Al-Kharj 11942, Saudi Arabia
2   Department of Civil Engineering, Canadian International College (CIC), 6th October, Giza 12577, Egypt
3   Department of Quantity Surveying, Federal University of Technology, Akure 340110, Nigeria
4   CIDB Centre of Excellence, Faculty of Engineering and Built Environment, University of Johannesburg, Johannesburg 2092, South Africa
5   School of Social Sciences, Universiti Sains Malaysia, Penang 11800, Malaysia
6   Construction and Building Engineering Department, College of Engineering and Technology, Arab Academy for Science, Technology and Maritime Transport (AASTMT), B 2401 Smart Village, Giza 12577, Egypt
7   Islamic Architecture Department, Faculty of Engineering & Islamic Architecture, UMM AL QURA University, Mecca 24382, Saudi Arabia
8   College of Science and Computer Engineering, Taibah University, Yanbu 46421, Saudi Arabia
*   Correspondence: a.farouk.kineber@gmail.com or a.kineber@psau.edu.sa

**Abstract:** Sustainability concepts should be adopted via new technologies to achieve the greatest possible gains without compromising the objectives of projects. In this research, we empirically investigated the influence of identified drivers on the implementation of robotics in the building sector of developing countries. To this end, with a view to sustainable building projects, the drivers of robotics were derived from the literature, which were subsequently contextually adjusted using a survey method through the exploratory factor analysis (EFA) method. The results of EFA revealed that the drivers of robotics can be classified into three primary constructs: technology, industry, and culture. However, the benefits of implementing robotics can be grouped into two primary constructs: resources and environment. Therefore, in this study, we employed partial least squares structural equation modeling (PLS-SEM) to evaluate the connections amongst drivers and applications of robotics in Nigeria's building industry. The results indicated that the input to the implementation of robotics in Nigeria's building industry via the drivers of robotics has a considerable influence at a small scale, with an impact of 14.5%. The findings of this study can serve as a guide for policymakers looking to improve their projects and increase sustainability by using robotics in the building sector.

**Keywords:** construction projects; sustainability; robotics; resources; environment; structural equation modeling

## 1. Introduction

The architectural, engineering, construction, and operations (AECO) industry is a basis of a country's economy, expected to accounting for approximately 15% of global GDP by 2030 [1,2]. The building industry is one of the key sectors of the economy that define the healthy lifestyle and well-being of the population of any country [3]. Building projects utilize approximately 40% of global power and is responsible for up to one-third of global greenhouse gas emissions in both rich and developing countries [4]. Given the growing concentration of people and economic activity in many cities, such projects are critical to the accomplishment of global sustainability goals [5]. Furthermore, increased

urbanization in these cities has resulted in an increase in urban populations [6]. Nonetheless, in a rapidly changing and urbanizing world, building allocation cannot keep up with demand [7]. Significant changes in the building industry have been recorded in many developing countries, driven by the need to attain national objectives [8]. It was reported that 828 million impoverished people in developing nations live in slums and substandard housing, and this population is expected to grow to 1.4 billion by 2020 [7,9,10]. These regions have experienced tremendous growth, with an emphasis on the importance of building to ensure a basic way of life [11]. Consequently, all governments have prioritized inexpensive building projects by enacting a variety of affordable building technologies and laws [3]. However, a debate continues regarding whether such structures are affordable for low-income earners [7]. In addition, the building industry in these countries is not competitive, as it cannot meet global requirements for sustainable development. Such projects are marred by multiple resource challenges, such as budget breaks, delays in scheduling, non-completion, high risk of failure, and poor quality, making it difficult to achieve the anticipated goals. These projects face multiple challenges [12,13]. Considering the low level of investment in this sector, many projects have been cancelled or suspended [14]. In all developing countries, the building industry meets the demands of the government demand, society, and consumers, consequently lagging other comparable businesses [15] Furthermore, the problem of sustainability in the low-income building industry has not been solved and considered [16]. As a result, the necessity of "sustainable buildings" that are ecologically friendly and resource-efficient has been stressed in the literature. Wolstenholme et al. [17] argue that implementing effective and sustainable building practices will revolutionize the building industry. Moreover, building stakeholders are unable to quantify the environmental impacts of structures as they are created [18]. As a result, robotics implementation may be integrated throughout the life cycle of a project through sustainability techniques [19,20].

The use of robotics in building has been acknowledged as one of the most radical advances for building projects [21]. Skibniewski [22] characterized the application of robotics in the construction industry as sophisticated construction equipment capable of being teleoperated, gathering and processing sensory data, and being numerically controlled for autonomous job implementation. According to Mahbub [23], robotics implementation entails the use of self-controlled mechanical and electrical gear with intelligent control mechanisms for building jobs and operations. Several building firms have already begun to employ modern construction technology to decrease waste and resource consumption [21]. The implementation of on-site robotics construction technologies has proven to provide several advantages, including considerable waste reduction, major time savings, flexible working conditions, and increased quality but at a high initial cost [24,25]. At the business level, long-term economic value in terms of payback duration and return on investment may be used to analyze the financial viability of investments in robotics deployment [26].

Although robotics have come to represent a popular instrument for solving building difficulties in a number of industrialized countries, particularly in terms of investigating the functions of computerized technologies to accelerate various sorts of construction markets, [27], most emerging countries have yet to seriously consider such technology. In the building industry of third-world nations, only an insignificant proportion of construction firms have successfully implemented robotics prefabrication in their projects. Notwithstanding the low implementation rate of robotics, there has been continuous research on innovative robotics and automation technologies incorporated into building sites. A groundbreaking technological advancement is the invention of 3D printing robots [28], which can be used to structurally print innocuous concrete buildings and bridges [29]. Humanoid robotics technology is (e.g., exoskeleton) another such invention, combining human intelligence with the speed, efficiency, and power of a robotics system fastened to the human body. Thus, it provides the capacity to manage constricted and complex tasks [30].

Apart from these innovative inventions, the building industry has not taken full advantage of the open possibilities offered by these technologies. Whereas many industries have

depended on and thoroughly explored the application of robotic technologies, such as the motorized medical profession, the universal application of robotics in the building sector is overdue as a result of many impediments [31]. The implementation of robotics in the building industry is quite sluggish. When generating different types of building jobs, the use of several single-task assembly robots is required [32], and robotics technology implementation is expected to enhance the sustainability of the building industry, as well as to solve issues such as labor shortages and safety hazards, particularly in high-rise buildings [33].

Kim et al. [34] argued that the low level of automation in the building sector delays other industries. An investigation involving 11 significant building firms and public organizations in Europe revealed that apart from the perception that automated and robotic systems can enhance output, safety, and health [35], there are substantial threats to the implementation of robotics. These include technical and commercial risks and the high cost of implementation [36]. However, some studies have attempted to classify drivers of adoption, especially for definite activities in the building industry; these experiments have primarily focused on global building firms instead of the building industry in third-world nations building, such as in Nigeria. The existing literature has not presented an all-inclusive analysis that recognizes the impediments to the conventional implementation of robotics in building activities within Nigeria's building industry. Consequently, filling this gap requires advanced knowledge concerning construction in third-world nations, especially Nigeria. To that end, we formulated the following research questions to be answered in this study. What are the benefits and impact of robotics in Nigeria's building industry? What are the requirements and drivers needed to implement robotics in Nigeria's building industry? Accordingly, this study was carried out to fill the existing gap by mathematically studying the connection concerning robotics drivers and adoption and the impact of implementing robotics in the building industry using the partial least squares modeling technique. We used the global–local context (GLC), which emphasizes the worldwide importance of the study subject. Furthermore, this strategy both represents and accentuates the issues under consideration. Thus, the method has been embraced by "developing" countries, including Nigeria, as the local environment is geared to produce such clarity (i.e., establishing its importance). The outputs from this study can be useful in enabling numerous attractive benefits for many building professionals, such as policymakers, project bidders, and architects [37], not only in Nigeria but also in other developing countries where building projects are carried out in the same way. This study will offer valuable insight that can help in decision making with respect to successful construction projects by enhancing building resources and improving the entire building environment through robotics implementation.

## 2. Model Development and Research Background

In 1921, Karel Capek coined the concept of robotics. At present, robots are utilized on farms, in workplaces, homes, streets, and other public places, such as restaurants, malls, and amusement parks, in the form of drones, humanoids, and self-driving vehicles. Robotics has been a promising and evolving field of science and engineering since the 1920s. Initially, robots were utilized for complex routinized tasks and limited to factories and warehouses, whereas they are currently an integral part of human society. At present, robots are applied for various purposes, such as in food manufacturing [38], therapeutic training [39], teaching [40], and drain cleanup [41]. The pace of automation is expanding very precipitously, which is apparent from the rate at which robots are applied in the construction industry. Robots have made significant advances in the socioeconomic perceptions of human society [42]. Consequently, the global manufacturing network paradigm is evolving, with increased use of artificial intelligence and robots enabling manufacturing intelligence and smart production [43].

Robots are grouped into two major classes: field and service robots. The latter are applied in domestic contexts, such as homes, public parks, and restaurants and are typically referred to as humanoids. Field robots are specific and unique robots designed to

operate in a specific milieu, such as ground, aerial, and marine applications [42]. The developments of robotics can be grouped into three generations. The first generation comprised robots primarily used for carrying out repeated tasks and for automation. Those dealing with weaponry, the development of tasks, entertainment, and research are termed second-generation robots. Those concerned with intelligence that can co-operate and coexist with humans constitute third-generation robots. These robots are primarily applied to understand patterns of behavior and natural languages and respond to human behavior [44–46].

The application of robotics offers active customer care and involvement through decision making. Rockart [47] described drivers as locales in which findings guarantee a company's commercial success if adequate. Chan et al. [48] and Yu et al. [49] concur that drivers can be deemed as essential for managing planning and field actions to guarantee achievement [50]. Based on Dillon and Morris' [51] definition of technology implementation, the operator's readiness to employ technology in day-to-day life to complete assignments specifies customer acceptance. Other studies, including [52], have examined the implementation of technology in everyday life. The analyzed theories typically see the drivers of technology implementation as achievement aspects [53–55].

The theory of reasoned action (TRA) is concerned with the elements of intentionally planned behaviors [56]. A task abstraction module (TAM) was pioneered to evaluate operator adoption of high-tech modernizations [57]. the unified theory of acceptance and use of technology (UTAUT) is, likewise, a well-established theory developed on the basis of TRA and TAM [52]. Additionally, precise methods and tools are needed to boost the implementation of robotics developed for building projects. Robotics constitute a technique that entails the application of approaches that proffer a motivation to achieve improved project and environmental resources. The speediest innovations in automation and robotics are driven by software programming methods that enable applications to find patterns, conduct analysis, and make predictions according to various sources of data. Many types of methods are used in nearly all industries to enhance the precision, quality, and pace of specialized processes.

The building industry is dominated by equipment and plants that generate considerable emissions. If robotic machinery is used in place of such equipment, it can reduce environmental pollution and create a friendly environment. This includes robots whose primary function may not affect the environment, although they might have considerable environmental consequences. Robots are designed to explore their effect on the environment, although they have substantial environmental effects. Robots are also designed to delve into new environments that are impenetrable by humans [58]. The desire for modernization in the building industry hinges on the willingness of the industry to adopt novel technologies. Implementing novel technologies tends to lessen some obstacles that can affect the building industry. Carmona [59] argued in favor of building technology, arguing that the intrinsically conservative nature of the sector has failed to transform construction processes. However, results have lagged, despite the availability of new methods that can reduce construction costs.

Consequently, the application of new building methods can aid in the expansion of the effectiveness of tasks and reduce the high costs devoted to building projects. Among the significant benefits of robotics, implementation improves working conditions by avoiding contact with dangerous tasks and reducing such tasks among workers. Robotics implementation can boost occupational safety by performing unsafe tasks in hazardous areas that would otherwise have to be performed by humans [60,61]. Robotics implementation has led to a reduction in injuries to workers and laborers. Issues stem from the quality of work delivered by employees and the repetitiveness of tasks performed, and labor costs can be reduced if fewer workers are required. Currently in the construction industry, human tasks are being supplemented with robots to cut down on the number of workers. Tables 1 and 2 show the benefits and drivers of robotics adoption, respectively. Likewise, Figure 1 shows the conceptual framework for the present study.

**Table 1.** Drivers leading the application of robotics for construction projects.

| Code | Driver | [62] | [63] | [64] | [65] | [66] | [67] | [68] | [69] | [70] | [71] | [72] | [73] |
|---|---|---|---|---|---|---|---|---|---|---|---|---|---|
| D1 | Technology innovation | | √ | | | | | | | √ | | √ | |
| D2 | Environmentally friendly nature of robots | | √ | | √ | | | | √ | √ | √ | | |
| D3 | The need for connectivity and convergence | | √ | √ | √ | | √ | | √ | | | | |
| D4 | The rapidly-changing, field- and project-based nature of the industry | | √ | √ | | √ | | √ | √ | | | | |
| D5 | Integration and globalization of the construction industry | √ | | √ | | | √ | | | | | | |
| D6 | The use of ICT in the construction industry | | | | | √ | √ | | | | √ | | √ |
| D7 | Rapid advancement in software programming technologies | | | | | √ | √ | | | | √ | | √ |
| D8 | Software visualization and artificial vision | | √ | | | √ | √ | | | | √ | | √ |
| D9 | Fusion of traditional and innovative technology | | √ | | | √ | √ | | | | √ | | |
| D10 | The need for urbanization | | √ | | | | | | | | | | |

**Table 2.** Benefits through the adoption of robotics for construction projects.

| Code | Benefit | [62] | [63] | [64] | [65] | [66] | [67] | [68] | [62] | [74] | [70] | [75] | [75] | [76] | [23] | [72] | [77] |
|---|---|---|---|---|---|---|---|---|---|---|---|---|---|---|---|---|---|
| RI 1 | Improves real-time planning and saves time | √ | | | | √ | √ | | | | √ | √ | | | | √ | √ |
| RI 2 | Better accuracy than that of site laborers | | √ | | √ | | √ | | √ | | | | | | | √ | |
| RI 3 | Enhances efficiency and improves the quality of work | | √ | | | | √ | | √ | √ | √ | √ | | | √ | √ | √ |
| RI 4 | High standards for health and safety measures | | √ | | | √ | | | √ | | √ | √ | √ | | | √ | |
| RI 5 | Reduces hazardous risk | | | | √ | | √ | √ | √ | √ | √ | √ | | | | | |
| RI 6 | Enhances existing construction plants and equipment | | | | | | √ | √ | √ | | | | | | | | |
| RI 7 | Decreases labor intensity | | | √ | | √ | | | | | √ | √ | | | √ | √ | |
| RI 8 | Reduces waste of building materials | | | | | | | | | | | | | | | | |
| RI 9 | Reduces dependence on direct labor | | | | | | | | √ | | | | | | | | |
| RI 10 | Makes construction plants and equipment more environmentally friendly | | | | √ | | | | | | | √ | | √ | | | |
| RI 11 | Improves the process of construction activities | | √ | | | | √ | | | | | | | | | | |
| RI 12 | Greater control over the productive process | | | | | √ | √ | | | | | | | | | | |

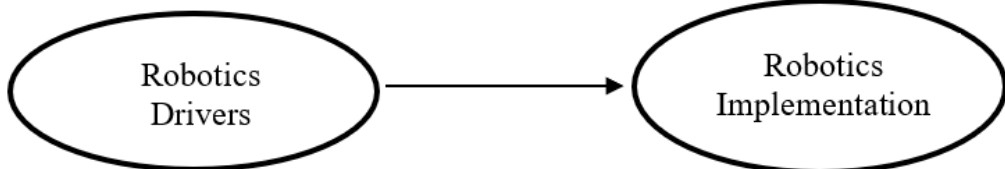

**Figure 1.** Impact of robotics drivers on robotics implementation.

### 3. Research Methodology

The study approach involves creating a theoretical model that summarizing a literature review used to create transitional theories (or hypotheses) that were verified by employing experimental proof [78]. A three-stage procedure was established for the conceptual modeling process: (i) identifying the model's construct, (ii) grouping the model's constructs, and (ii) itemizing the connections between the model's constructs [79]. The model's results were obtained following this process, as shown in Figure 1. The study plan was based on the work of Tanko et al. [80], as illustrated in Figure 2. Because the use of robotics is comparatively new in Nigeria, in this study, we employed a stratified sampling approach to attain a particular subpopulation group [81]. This method was proposed to help researchers collect highly reliable and accurate data because the current survey is associated with a topic concerning robotics. The benefits of stratified sampling, as highlighted by Sharma [82], are as follows: (i) "Decreasing bias in sample case selection, this also implies that perhaps the sample will represent a substantial portion of the surveyed populations; (ii) Allow the sample to be generalized to the population. The population difference is considered by stratification, along with all three sectors (client, contractor, and consultant) and most five subsectors in Nigeria" [83]. Respondents reported robotics drivers and adoption benefits on an experience and knowledge basis utilizing a five-point Likert scale, with scores of 5 and 1 representing very high and very low, respectively, with high, average, and low scores falling between 5 and 1. This scoring system has been broadly applied in many studies, including those concerned with construction management [84–90]. The study was designed to provide stakeholders with a variety of solutions based on practice in a range of construction schemes.

The size of the sample was determined according to [91]. More than thirty (30) cases were considered adequate for further examination, including mean, median, and mode for a normal distribution curve [92]. In contrast, Harris and Schaubroeck [93] argued that a sample size of at least 200 is required to warrant vigorous SEM. A very intricate path model requires a sampling size of 200 or more, as suggested by Kline [94], whereas a sampling size greater than 100 was considered satisfactory by Yin [95]. Because in this analysis, we employed the SEM method, 104 participants were recruited among 180 construction experts. The participants were personally recruited and completed a self-administered questionnaire for SEM evaluation, with a response rate of approximately 68%, which was deemed satisfactory for this study [96,97].

#### 3.1. Exploratory Factor Analysis

Exploratory factor analysis (EFA) was conducted to examine the groups mentioned above based on a questionnaire distributed to experts in Nigeria's construction industry. Between 150 and 300 samples or observations are required for EFA [98]. However, Pallant [99] argued that the investigators have some leeway concerning the sampling size for factor analysis. Thus, the use of larger sampling sizes is advocated relative to the number of variables involved. Shen [100] suggested that a set of 20 to 50 variables or parameters is appropriate for factor analysis. Because the individual aspects are not sufficiently determined if the quantity of variables exceeds this limit, some analyses required fewer variables if the sampling size is sufficiently large [14,101]. The population included in the present study was considered a good representative sample across the relevant ranges [98]. Accordingly, the ten identified variables and the completed questionnaires acquired from 104 respon-

dents in the current analysis were, likewise, deemed suitable for factor analysis [98,102].

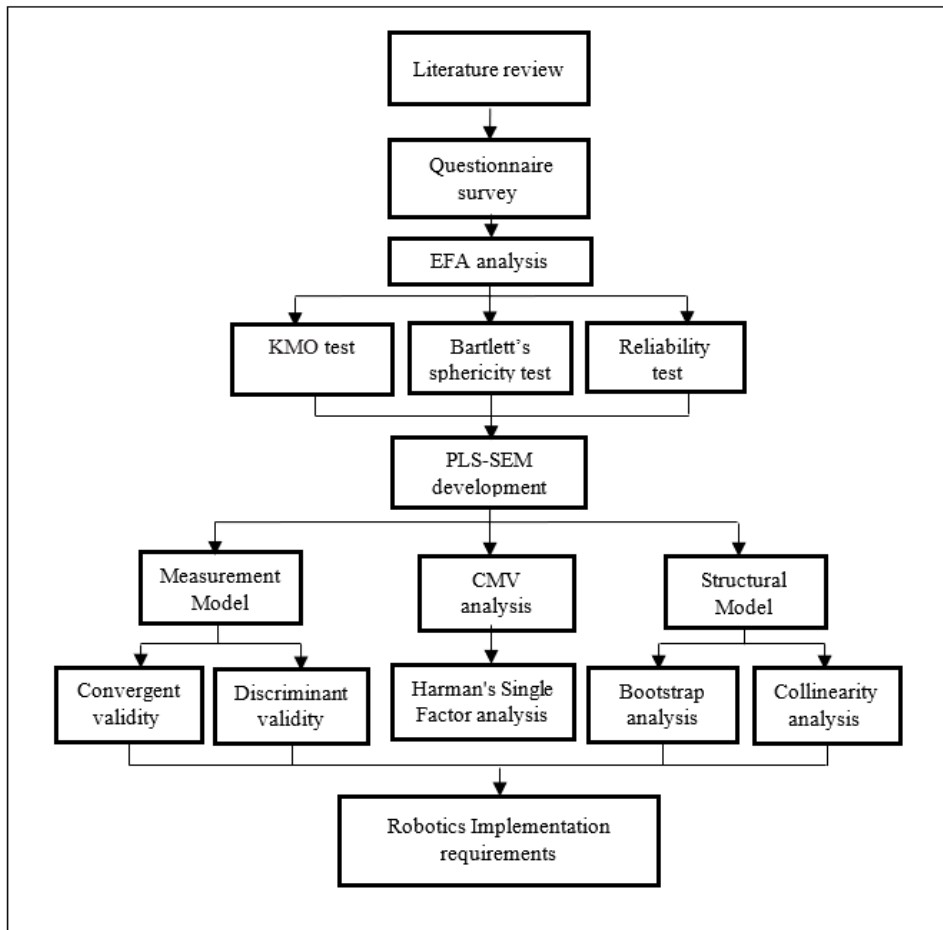

**Figure 2.** Research design.

### 3.2. Analytical Method (Structural Equation Modeling)

To examine the implementation of robotics, we conducted a literature review, and four models were compared in order to develop an optimal model of robotics implementation that guarantees practical construction projects. The considered models were multiple linear regression (MLR), structural equation modeling (SEM), artificial neural network (ANNs) and system dynamics (SD). Owing to the connections among non-observed variables, a regression model was not applied. This is a major limiting factor for the application of regression model [103]. Likewise, SD could not be applied because the nature of the survey data is unconnected to the period. ANNs are also a projecting tool, and the purpose of the analysis is to analyze the effects of the drivers of robotics adoption on robotics implementation. The SEM technique explains the relationship between many quantifiable and non-observable variables, making it appropriate for the analysis in this study [104]. Amaratunga et al. [105] explained that SEM is a valuable instrument for tackling errors within variables. This SEM analysis technique was used to generate a model and determine the relationships among robotics drivers and robotics implementation. Byrne [73] noted that SEM is a widely recognized non-experimental investigation used for parameter evaluation and hypothesis testing [106]. Similarly, Ringle et al. [107] confirmed that this technique has been improved over many decades in a research journal bulletin issued in MIS (Management Information Systems) Quarterly. Furthermore, Yuan et al. [108] concluded that SEM is a thoroughly understood and prevalent form of data analysis in the social sciences.

The SEM technique was used in the current analysis because it has been widely applied in studies concerning the construction industry [109]. This approach enabled us to test

hypothesized relationships simultaneously [110]. In order to establish the relationships between robotics drivers and robotics implementation, we used the partial least squares model (PLS) [111], which includes both formative and reflective variables. It enables examination of the drivers within robotics and the effects of robotics implementation. PLS-SEM, on the other hand, allows for the comprehensive testing of complicated models for their similarity with the data, as well as the testing of explicit assumptions about parameters for their similarity with the data [112]. In the PLS, the computing model defines the relationsships among the constructs (i.e., robotics drivers) and the experiential pointers (or drivers) [113,114]. In this study, the goal of the data reduction procedure was to decrease the number of variables and parameters in the research model to a reasonable amount relative to the sample size/parameters estimated in the SEM ratio [115].

## 4. Results

### 4.1. Characteristics of the Respondents

Data on the characteristics of the respondents were collected for background information, including the profession of the respondent, professional qualifications, academic qualifications, and years of professional experience. Table 3 shows the profession of the respondents who completed the questionnaire; a proportion of 28.8% of respondents were architects, 28.8% were quantity surveyors, 11.5% were builders, 25% wre engineers, and 5.8% were project managers. Table 3 shows the years of professional experience of the respondents; a proprotion of 32.7% of respondents had fewer than 5 years of professional experience, 32.7% had 6 to 10 years of professional experience, 21.2% had 11 to 15 years of professional experience, 11.5% had 16 to 20 years of professional experience, and 1.9% had more than 21 years of professional experience. Table 3 shows the knowledge level of the respondents about robotics, with three possible responses; a proportion of 73.1% of respondents anwered 'Yes' to this question about knowledge of robotics, 21.2% answered 'No', and 5.8% answered 'Maybe' indicating that they were not sure about the answer to the question. Table 3 shows the knowledge level of the respondents about the use of robotics in the construction industry, with three possible responses; a prorportion of 71.2% of the respondents answered 'Yes', 21.2% answered 'No', and while 7.7% answered 'Maybe' indicating that they were not sure about the answer to the question.

### 4.2. Identifying and Categorizing the Model's Constructs

A total of 20 elements pertinent to robotics drivers and 12 items regarding the benefits of robotics implementation were examined using EFA. Many recognized factorability parameters were employed for model construction. The KMO is a homogeneity of factor dimension widely applied to assess whether partial associations between items are negligible [116–118]. The KMO index must be between 0 and 1 for effective factor analysis and should have a value of at least 0.6 [98]. Bartlett's sphericity test also indicates whether the correlation matrix is identical. An appropriate factor analysis requires Bartlett's sphericity test, as suggested by Pallant [119]. A *p*-value <0.05 is considered significant [120]. The initial results for robotics drivers and benefits of robotics implementation show that the Kaiser–Meyer–Olkin sample suitability ratios were 0.730. and 0.882, respectively. These values are greater than the recommended Bartlett's value of the sphericity test. It was significant for robotics drivers ($x^2(45) = 123.511$, $p < 0.05$) and for benefits of robotics implementation ($x^2 (66) = 270.008$, $p < 0.05$). Moreover, the entire diagonal of the anti-image's correlation matrix are above 0.5 and putative, given the insertion of individual elements in the factor analysis. The preliminary communalities are estimations of variance within variables measured among all components, and lower values (<0.3) designate variables that do not fit the solution factor. In the current analysis, all initial communalities were greater than the threshold, and all factor loadings were above 0.

**Table 3.** Characteristics of the respondents.

| | Frequency | Per Cent | Cumulative per Cent |
|---|---|---|---|
| Architect | 15 | 28.8 | 28.8 |
| Quantity Surveying | 15 | 28.8 | 57.7 |
| Builder | 6 | 11.5 | 69.2 |
| Engineer | 13 | 25.0 | 94.2 |
| Project Manager | 3 | 5.8 | 100.0 |
| Total | 52 | 100.0 | |
| Professional qualification of respondents | | | |
| Less Than 5 Years | 17 | 32.7 | 32.7 |
| 6 To 10 Years | 17 | 32.7 | 32.7 |
| 11 To 15 Years | 11 | 21.2 | 65.4 |
| 16 To 20 Years | 6 | 11.5 | 86.5 |
| Above 21 | 1 | 1.9 | 98.1 |
| Total | 52 | 100.0 | 100.0 |
| Knowledge Level About Robotics | | | |
| Yes | 38 | 73.1 | 73.1 |
| No | 11 | 21.2 | 94.2 |
| Maybe | 3 | 5.8 | 100.0 |
| Total | 52 | 100.0 | |
| Knowledge Level About use of Robotics | | | |
| | Frequency | Per cent | Cumulative per cent |
| Yes | 37 | 71.2 | 71.2 |
| No | 11 | 21.2 | 92.3 |
| Maybe | 4 | 7.7 | 100.0 |
| Total | 52 | 100.0 | |

The results of exploratory factor analysis for all ten elements relevant to robotics drivers indicated two factors with eigenvalues or more than 1. The total variance and the eigenvalues described by the two factors were 57.915%. It is noteworthy that D 1 was excluded from the primary analysis, owing to cross loading, as shown in Table 4. Additionally, the EFA results revealed that all 12 elements were relevant to the robotics implementation benefits with three extracted. EFA results also identified ten elements relevant to the robotics derivers, with three extracted factors with eigenvalues of more than 1. The total variance and the eigenvalues described by the three factors were 59.88%. However, as shown in Tables 4 and 5, three cross-loading factors (RI 5, RI 8, and RI 9) were excluded from the main study.

**Table 4.** Factor loadings of robotics drivers.

| Driver | Components | | |
|---|---|---|---|
| | 1 | 2 | 3 |
| D 1 | 0.488 | | 0.587 |
| D 2 | | 0.642 | - |
| D 3 | 0.657 | | - |
| D 4 | | 0.692 | - |
| D 5 | 0.721 | | - |
| D 6 | - | 0.784 | - |
| D 7 | - | - | 0.867 |
| D 8 | - | - | 0.684 |
| D 9 | 0.651 | - | - |
| D 10 | 0.763 | - | - |

**Table 5.** Factor loadings of robotics benefits.

| Benefit | Components | |
|---|---|---|
| | 1 | 2 |
| RI 1 | 0.546 | |
| RI 2 | - | 0.650 |
| RI 3 | - | 0.791 |
| RI 4 | - | 0.740 |
| RI 5 | 0.609 | 0.564 |
| RI 6 | - | 0.669 |
| RI 7 | 0.683 | - |
| RI 8 | 0.523 | 0.550 |
| RI 9 | 0.507 | 0.605 |
| RI 10 | 0.821 | - |
| RI 11 | 0.766 | - |
| RI 12 | 0.743 | - |

For the factors extracted by EFA, statistical reliability was determined. Factors for individual phases of the factor (or group) were measured based on the maximum factor loading for each parameter in the matrix structure. The table also shows that the reliability test was satisfactory. According to Nunnally [121], Cronbach alpha value should be higher than 0.6 for recently developed dimensions, although the expected value was 0.7, and values greater than 0.8 were deemed reliable. Therefore, the total Cronbach values were satisfactory, as they were higher than 0.6, and usual correlations of variables were above 0.3 for all factors, suggesting harmonious inner variables [122].

*4.3. Common Method Bias*

Calculation of variance errors that influence the validity of samples is referred to as general process bias, i.e., the statistical error variance for the observed and expected variables [123]. It is calculated using Harman's single-factor model, which suggests different construction measures [124]. In this analysis, a single-factor test was used for variance calculation [125]. If the overall variance of the variables is less than 50%, the traditional method bias has a minimal impact on the results [124,126]. "When such a single-factor test is conducted, a high common method variance may exist if all factors come down to one single factor or a special factor accounts for the majority of total covariance over all variables" [127]. As shown in Table 6, the first set of variables accounts for 47.9% of the overall variance, indicating that the common method variance cannot be influenced by less than 50% [124].

**Table 6.** Common method variance.

| Sum of Squared Loadings | | |
|---|---|---|
| Total | % of Variance | Cumulative % |
| 9.11 | 47.9 | 47.9 |

*4.4. First-Order Construct Measurement Model*

The SEM shown in Figure 3 duplicates the theoretical study model presented in Figure 1. As shown in Tables 1 and 2, the model constructs of each robotics driver and implementation benefits model were comprehensive and categorized based on elements obtained from the literature. According to Hair Jr et al. [128], the evaluation model requires an estimate of (i) an indicator's reliability, (ii) merged reliability, (iii) extracted average variance, and (iv) discriminant ability. In the current analysis, the PLS algorithm was applied 300 iterations using the following settings [129] recommended by Wong [130]: weighing scheme, weighing path, data matrix with a mean 0, variance of 1, highest interactions of 300, abort criterion of $1.0 \times 10^{-5}$, and initial weights of 1.0.

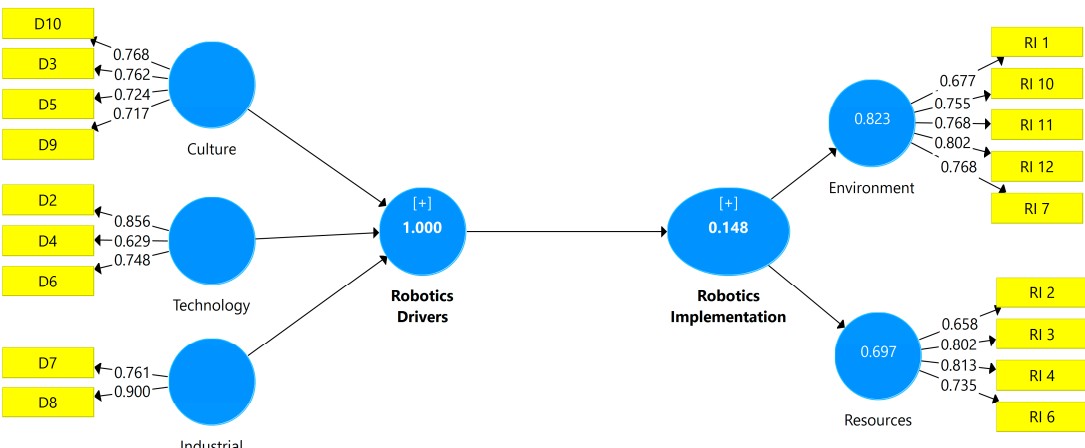

**Figure 3.** SEM model with R2 values and path coefficients.

Typically, indicators with external loadings in the range of 0.40 to 0.70 must be considered for elimination if removal of the indicator results in a significant increase in the reliability of the composite and AVE [131]. External load variables below 0.60 were found to disagree with this prerequisite and were eliminated for further analysis, as suggested in [128]. At this level, roughly one-half of an indicator's variance is described by its components. The intensity at which the variance was described is higher than the variance error. Figure 3 and Table 4 show the external loadings for all variables of the measurement model. Thus, all external loads were greater than 0.60, which was deemed acceptable. Because the Cronbach alpha limits computational sensitivity with respect to the number of variables considered, the core constancy of composite reliability (*cr*) was evaluated according to Hair Jr et al. [128]; values greater than 0.70 were considered acceptable. For exploratory research, values greater than 0.60 are deemed suitable [130]. All models met the *cr* threshold of >0.70 and were therefore accepted, as summarized in Table 7. The AVE is a technique that is commonly used to estimate the convergent cogency of constructs within models with values higher than 0.50 [132,133], suggesting an acceptable value as recommended in [130]. All constructs passed this test, as shown in Table 7.

**Table 7.** The results of convergent validity.

| Construct | Item | Outer Loading | Cronbach's Alpha | Composite Reliability | AVE |
|---|---|---|---|---|---|
| Culture | D10 | 0.724 | 0.730 | 0.831 | 0.552 |
| | D3 | 0.717 | | | |
| | D5 | 0.762 | | | |
| | D9 | 0.768 | | | |
| Technology | D2 | 0.658 | 0.606 | 0.792 | 0.562 |
| | D4 | 0.813 | | | |
| | D6 | 0.735 | | | |
| Environment | RI 1 | 0.677 | 0.811 | 0.869 | 0.570 |
| | RI 10 | 0.755 | | | |
| | RI 11 | 0.768 | | | |
| | RI 12 | 0.802 | | | |
| | RI 7 | 0.768 | | | |

**Table 7.** *Cont.*

| Construct | Item | Outer Loading | Cronbach's Alpha | Composite Reliability | AVE |
|---|---|---|---|---|---|
| Resources | RI 2 | 0.658 | 0.746 | 0.840 | 0.569 |
| | RI 3 | 0.802 | | | |
| | RI 4 | 0.813 | | | |
| | RI 6 | 0.735 | | | |

For constructs differing considerably from other constructs, discriminant analysis was accurately defined based on the observed standards. Consequently, the establishment of discriminative validity (DV) indicates that a construct is typical and describes singularities that are not clearly defined by the remaining constructs within the model [134]. DV can be computed by applying two types of methods: the cross-loading criterion and Fornell and Larcker's (1981) criterion. The square root of the AVE of the individual construct in the initial technique can be equated to the correlation of an individual construct with any other construct to estimate discriminant validity. The square root of the AVE should be greater than the correlation among the dormant parameters based on the Fornell and Larcker [135] values. The results confirming the discriminant validity of the measurement model are presented in Table 8 [136].

**Table 8.** Correlation of latent variables and discriminant validity (Fornell–Larcker).

| | Culture | Environment | Industry | Resources | Technology |
|---|---|---|---|---|---|
| Culture | 0.743 | - | - | - | - |
| Environment | 0.383 | 0.755 | - | - | - |
| Industry | 0.379 | 0.248 | 0.834 | - | - |
| Resources | 0.26 | 0.527 | 0.223 | 0.754 | - |
| Technology | 0.455 | 0.173 | 0.303 | 0.127 | 0.75 |

The cross-loading criterion was employed to measure discriminative validity. This technique represents an attempt to establish the loadings of indicators based on the assumed dormant construct, which must be greater than the loadings on the remaining construct in a rows, which means that the indicator's (items) loadings for each construct must be greater than the loadings of alternative constructs. Table 9 shows that the loading of each point of the allotted dormant construct is greater than the cross loading on the alternative constructs in a row.

*4.5. Measurement Model (Second-Order Construct)*

The significant variables (dependent and independent variables) were categorized as second-order static variables, and the momentous input of all first-order dormant variables was then examined by employing the bootstrap method. The individual construct of the robotics drivers was determinative, and robotics implementation was found to be an insightful construct. Significant correlations among the formative indicators of the measurements model were not characteristically projected. Furthermore, significant correlations among the formative variables indicating collinearity is deemed awkward [134]. Analysis of the value of the variable inflation factor (VIF) revealed collinearity between the construct's formative variables. For this assessment, the internal VIF values were used to evaluate collinearity problems related to the formative–reflective form of second-order constructs. The three (3) first-order subscales of robotics drivers comprised culture, technology, and industry. For culture, maximum external loading was observed ($\beta$ = 0.621, $p < 0.001$), as indicated in Table 10, followed by industry ($\beta$ = 0.277, $p < 0.001$) and technology ($\beta$ = 0.369, $p < 0.001$). According to these findings, all VIF values were less than 3.5, signifying that those subdomains contributed individually to higher-order constructs.

**Table 9.** Cross-loading testing of the discriminant validity of indicators.

| Item | Culture | Technology | Industry | Environment | Resources |
|---|---|---|---|---|---|
| D10 | 0.768 | 0.347 | 0.22 | 0.198 | −0.103 |
| D3 | 0.762 | 0.433 | 0.346 | 0.358 | 0.371 |
| D5 | 0.724 | 0.245 | 0.313 | 0.287 | 0.194 |
| D9 | 0.717 | 0.313 | 0.243 | 0.288 | 0.293 |
| D2 | 0.472 | 0.856 | 0.216 | 0.121 | 0.124 |
| D4 | 0.263 | 0.629 | 0.19 | 0.079 | 0.236 |
| D6 | 0.252 | 0.748 | 0.283 | 0.193 | −0.066 |
| D7 | 0.185 | 0.205 | 0.761 | 0.186 | 0.277 |
| D8 | 0.411 | 0.29 | 0.9 | 0.225 | 0.128 |
| RI 1 | 0.314 | 0.205 | 0.393 | 0.677 | 0.388 |
| RI 10 | 0.225 | 0.005 | 0.115 | 0.755 | 0.302 |
| RI 11 | 0.258 | 0.071 | 0.157 | 0.768 | 0.416 |
| RI 12 | 0.332 | 0.193 | 0.141 | 0.802 | 0.443 |
| RI 7 | 0.314 | 0.18 | 0.148 | 0.768 | 0.429 |
| RI 2 | 0.049 | 0.028 | 0.161 | 0.305 | 0.658 |
| RI 3 | 0.272 | 0.318 | 0.32 | 0.412 | 0.802 |
| RI 4 | 0.182 | 0.003 | 0.226 | 0.478 | 0.813 |
| RI 6 | 0.259 | 0.074 | 0.216 | 0.376 | 0.735 |

**Table 10.** Second-order model testing by means of bootstrapping for foundational constructs.

| Path | β | SE | T | *p*-Value | VIF |
|---|---|---|---|---|---|
| Culture → Drivers | 0.621 | 0.054 | 11.499 | <0.001 | 1.373 |
| Industrial → Drivers | 0.277 | 0.05 | 5.588 | <0.001 | 1.19 |
| Technology → Drivers | 0.369 | 0.053 | 7.003 | <0.001 | 1.294 |

Robotics implementation was identified as a second-order construct within the re model with two subsamples, comprising two components, i.e., environment ($\beta = 0.907$, $p < 0.001$) and resources ($\beta = 0.907$, $p < 0.001$), which contributed considerably to robotics implementation as a static second-order variable; standardized coefficient paths (external loadings) were greater than 0.7 and statistically significant, as indicated by Table 11.

**Table 11.** Second-order model testing by means of bootstrapping for weighty second-order constructs.

| Path | B | SE | T | *p*-Value |
|---|---|---|---|---|
| Robotics Implementation → Environment | 0.907 | 0.018 | 51.39 | <0.001 |
| Robotics Implementation → Resources | 0. 907 | 0.033 | 25.674 | <0.001 |

*4.6. Path Analysis: Structural Model*

Path analysis (PA) is a linear statistical technique that is ideal for management and social sciences. Likewise, PA is an essential tool for concurrent examination of multifaceted associations [98]. Primary phase analysis requires the application of SEM. This model is applicable for the evaluation of associations among studied concepts. After model fitting, SEM is the subsequent primary phase within SEM analysis. SEM can be used to detect relationships among variables. In SEM, the connections between factors are described in detail. The connections among exogenous (or dependent) and independent variables are shown by the data [137,138]. SEM evaluation is based on the model's total fit, with theorized variable estimations trailed by importance, size, and direction [137]. The last component involves confirmation of the proposed analytical relationship on the basis of the research hypotheses presented in Figure 1.

SEM was applied to the research hypothesis. The effect of robotics drivers on robotics implementation was studied using PLS-SEM based on the study framework. The related research model hypothesis is illustrated in Figure 4. Within the context of the bootstrapping methodology, the consequence of the model's hypothesis was estimated. Arbitrary

resampling of the fundamental dataset comprised the process of bootstrapping to produce new samples of comparable size to the basic set of data. This approach tests the consistency of datasets and their statistical implications and, consequently, the error of the computed coefficient's path [139]. The standardized coefficient path (β) and *p*-values, as well as the significance of the pathway, are presented in Figure 4. Table 12 presents the results of the bootstrapping method, indicating the *p*-values of the model path. According to the results, the effects of robotics drivers on robotics implementation were significant and positive (*p* = <0.001, β = 0.384).

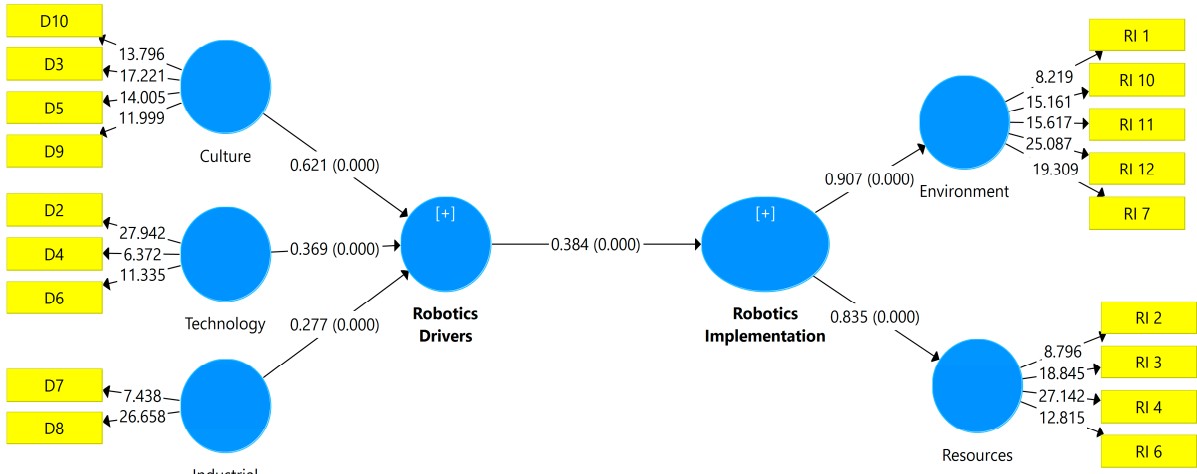

**Figure 4.** Bootstrapping analysis.

**Table 12.** Relative path for the model.

| Path | B | SE | T | *p*-Value |
|---|---|---|---|---|
| Drivers → Robotics Implementation | 0.384 | 0.083 | 4.631 | <0.001 |

### 4.7. Exploratory Supremacy of the SEM Model

The model results reveal robust reliability of each item, as well as the discriminatory and cogent validity of the measurement model. Besides calculating the variance within the dependent variable, which the model can define, the exploratory supremacy of the SEM model can be tested. Concerning the dependent variables within the model, the PLS algorithm reinforced squared multiple ($R^2$) relationships. The $R^2$ established by the PLS algorithm is comparable to the standard regression [140].

The value of $R^2$ signifies the sum of variance and can be explained using the independent variable within dependent variables. Consequently, higher $R^2$ values increase the analytical capacity of the SEM model. In the current analysis, the $R^2$ values were calculated by employing the Smart-PLS algorithm, as shown in Table 13. The adjusted $R^2$ for robotics implementation, as the significant dependent variable within the model, was 0.139, suggesting that the independent (or exogeneous) static variable (robotics drivers) can explain 13.9% of robotics implementation. These results imply that the effect of robotics drivers is insignificant, as argued by Chin [93].

**Table 13.** Coefficient of determination ($R^2$).

| Endogenic Dormant Variable | $R^2$ | Adjusted $R^2$ |
|---|---|---|
| Robotics Implementation | 0.148 | 0.139 |

### 4.8. Predictive Relevance of the Structural Model

A significant component of the proposed model is its capacity to assess analytical significance. A blindfolding protocol was employed for individual dependent variables to

assess the redundancy measures of cross validation. The results showed that project accomplishment with values of $Q^2$ (0.057) was greater than 0, suggesting that the autonomous construct has predictive relevance for the supported construct analyzed in this study [141]. The $Q^2$ value is greater than zero, as shown in Table 14. Thus, it can be assumed that the model has exceptional predictive significance.

**Table 14.** Predictive relevance ($Q^2$).

| Endogenic Dormant Variable | SSO | SSE$ | $Q^2$ (=1 − SSE/SSO) |
|:---:|:---:|:---:|:---:|
| Robotics Implementation | 936 | 882.36 | 0.057 |

## 5. Discussion

A wide range of equipment is used on construction sites [142]. However, "the selection of a variant of a life cycle of a building among a vast number of alternatives is an important problem in project management" [143]. Overall, there are fewer aspects within the construction industry than in other industries, comprising aspects such as productivity, quality, and product functions [144], and firms operate in a market that is becoming more complicated and dynamic in the current epoch and global environment [145]. Nevertheless, sustainable construction projects effectively promote environmental sustainability and social development. Implementation of robotics technologies among experts and meaningful actions can substantially improve the realization of construction projects. Automated systems and robotics have transformative potential and offer numerous benefits to the building, architecture, engineering, and construction industries [146,147]. A comprehensive foundation for identification of relationships among robotics drivers and their benefits in the proposed model was incorporated in an SEM analysis, in addition to statistics generated by model evaluations. As a result of revision and analysis, some fascinating results were discovered.

EFA analysis shows that the drivers of robotics implementation can be categorized into three main categories (culture, technology, and industry). The PLS-SEM results show that the greatest impact of these categorizes on robotics drivers was derived from cultural drivers, with an external path of 0.63, followed by technology and industrial drivers, with external paths of 0.399 and 0.277, respectively. EFA results likewise indicate that the implantation of robotics can be grouped into two significant groups: environment and resources, with external paths of 0.90 and 0.86, respectively. The independent and dependent variables were analyzed revealing the effect of robotics drivers and robotics implementation. The findings show that the robotics drivers contribute approximately 14.0% to implementation of robotics in the building industry.

Robotics drivers also exhibit a significant correlation with robotics implementation, with a β value of 0.384, achieving significance once an organization or a company implements one component of robotics drivers. Furthermore, robotics technology is improved by 0. 384 due to environmental and resources elements. The results show that the implementation of some robotics drivers supports the people engaged in implementing robotics technology for projects with the goal of sustaining the client's resources and meeting environment obligations.

The environmental aspect was found to be the most important, with an exterior loading score of 0.79. "Environmental benefits arise from enhanced rational use and reduction of the extraction of natural resources, reduction of water and energy consumption, conscious and orderly development" [148]. It has been argued that robotic systems can enhance the construction environment by reducing fatalities and liberating employees from performing hazardous assignments [16]. In this regard, the implementation of robotics is a possible solution to expand environmental resources and sustainability in many ways, including by reducing construction waste, saving natural resources, improving the safety of the workplace, and supporting an improved living atmosphere [21]. Furthermore, participants must reflect on how risks are reduced by robotics in construction firms, which can justify the high initial cost of investing and improving the construction environment [149].

To enhance the construction environment in the design stage, diverse automation design software and tools can be applied, in combination with parametric mechanisms, such as basic 2D drawing kits with parametric regulations via a wholly integrated 3D AutoCAD interface [23]. However, the formation of design concepts must be entrusted to an individual. Computers can provide considerable support through storage capacity and the ability to evaluate and maintain highly integrated and complex data designs [150], with tasks are carried out much faster compared to manual approaches, providing consistent and predictable productivity that reduces management oversight.

The performance of resources is described as a mixture of features needed for services required by the building project stakeholders and a foundation for the assessment of fitness and user satisfaction [151]. Resources factors had an exterior loading of 0.861, which was deemed reasonable. Results indicate that it is important to identify the components necessary for a project's success, which is one of the conventional benefits of implementing robotics technologies. These results are in line with those reported by Mao et al. [152], who argued that an autonomous classification robot based on image recognition might aid in the reduction in the amount of labor required for recycling activities. Pradhananga et al. [102] also posited that robotics implementation has many economic benefits and enhances the speed, quality, and productivity of construction. Additionally, robotic implementation has the potential to address some of the challenges associated with construction. Robotics might help to increase off-site construction, resulting in triple bottom-line benefits (economic, environmental, and social) by enabling more accurate building, closing the gap between intended and real energy usage [153].

Automation and construction robots are divided into three classes based on IAARC: those that improve existing construction equipment and factories, enthusiastic robots, and comparatively less cognitive (or intelligent) robots. For instance, the Obayashi Corporation's ABCs scheme improved construction schedules for forty (40)-story buildings within six months, and its Big Canopy system reduced the number of on-site workers required for concrete-strengthened structures by 75% [154]. Furthermore, the implementation of robotics can enable effective prefabrication, delivery, and supply of components to be undertaken according to the project schedule [155–158]. Automation implementation can also help to increase the coordination of planning throughout the project life cycle in terms of design, manufacturing, transportation, and installation, which are generally regards as difficult tasks by contractors due to the nature of the building industry, which is fragmented and varied and involves many parties [155,157]. The improved predictability and production quality associated with the adoption of engineering robots have led to an intensification of margins [159].

Based on the results reported above, we can conclude that robotics drivers will impact the success of the implementation of robotics technology under the influence of environmental and resource considerations. The obtained results concerning the achievement of robotics technology implementation through robotics drivers confirmed our study hypothesis. Therefore, the study objective was achieved. Our results also corroborate the existing literature, which indicates that environmental and resource considerations impact the implementation of robotics and that these variables influence the success of a project [160].

## 6. Conclusions

The construction industry is highly dependent on robotics in various countries, although this dependence is uncertain in third-world nations. Nigeria had experienced numerous loopholes and construction quality incongruities involving large-scale projects. This is typical of many developing countries. To improve these conditions, robotics technologies need to be implemented. To confirm the relationships between drivers of robotics and robotics implementation constructs, a PLS-SEM method was adopted.

Based on information gathered from 104 building experts, one direct and eight indirect paths were authenticated as essential for the development of a structural model. Moreover, the connections among factors via indirect and direct paths amongst drivers and robotics

implementation indicators and components were verified. The obtained results show that the adoption of robotics as mediated by the investigated drivers (proposed in the model) can enhance the sustainability of construction projects in terms of environmental and resource considerations. Significant findings of this study and its practical implications are as follows:

- Previous studies focused less on activities relating to robotics adoption and it is drivers. In this analysis, we narrowed the existing gap by examining the connection between robotics drivers and the adoption of robotics. The reported finding contribute significantly to the existing literature with respect to the management of building engineering by advancing the state of the art in robotics methods and requirements. Furthermore, the present study can serve as a basis for future research using analytically confirmed approaches concerning the implementation of robotics drivers, which a powerful and positive impact on robotics implementation in the construction industry. Moreover, the current research can inspire additional research works concerning construction projects.

- Although Nigeria is classified as a third-world nation with high population growth, most of its inhabitants have lower and medium income levels. There is a dire need for additional studies on the construction industry, owing to low environmental performance, especially concerning robotics drivers and robotics implementation. Such studies can inspire additional investigations on robotics drivers and implementation both within Nigeria and its neighboring countries. The results reported herein lay a foundation for robotics implementation both within and outside Nigeria. Our results can support the Nigerian construction industry by offering ways to reduce project costs and boost project resources.

- This analysis has many implications for experts in the construction industry, as well as contractors and project owners seeking to ensure the success of their projects by adopting robotics. The results reported herein can help stakeholders to adopt robotics by emphasizing the goal of the project with respect to resource and environmental requirements, which can impact the magnitude of the project's success.

## 7. Limitations and Future Research

We believe that research is a journey and not a destination. Therefore, we recommend research in the following areas:

1. In this research, we investigated the application of robotics for construction projects in Nigeria. Further research can be carried out by implementing robotics in construction projects.
2. Assessment of the effect of robotics on the performance of students in higher education institutions in Nigeria.
3. Current applications of building automation in Nigeria should be identified.

Based on the results of our study on the benefit of robotics in the construction industry, we propose the following recommendations:

1. Higher education institutions should train student on the application of new technologies and their applicability to construction projects. This will help to bring about enhanced student knowledge, which can be implemented after graduation.
2. Construction firms should educate construction professionals, such as architects, quantity surveyors, builders, project managers, and engineers on the importance and advantages of the application of robotics in the construction industry, and training should be organized.
3. Members should also be encouraged to adopt robotics in the construction industry so as to catch up with the ongoing trend in the construction industry and maintain relevance.
4. Suppliers should ensure that the cost of purchasing robotics equipment is reduced so that it can be easily accessible by various stakeholder in the construction industry.

**Author Contributions:** Research Idea: A.F.K.; Conceptualization, A.F.K. and A.E.O.; methodology, A.F.K. and A.E.O.; software, A.F.K.; validation, All Authors, formal analysis, A.F.K.; investigation, A.F.K. and A.E.O.; resources, All authors; data curation, All authors.; writing—original draft preparation, and A.F.K. and A.E.O.; writing—review and editing, All authors.; visualization, All authors.; supervision, A.F.K. and A.E.O.; project administration, A.F.K. All authors have read and agreed to the published version of the manuscript.

**Funding:** This study is supported via funding from Prince sattam bin Abdulaziz University project number (PSAU/2023/R/1444).

**Institutional Review Board Statement:** Not applicable.

**Informed Consent Statement:** Not applicable.

**Data Availability Statement:** Not applicable.

**Conflicts of Interest:** The authors declare no conflict of interest.

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
