# Peer review of "A Partial Least Squares Structural Equation Modeling of Robotics Implementation for Sustainable Building Projects: A Case in Nigeria"

_sustainability, doi:10.3390/su15010604_

Round 1

Reviewer 1 Report

The paper presents a general overview on drivers and benefits of implementing robotics in building industry in Nigeria. However, the title does not reflect the geographical limitation of the study. Generalizing the findings is questionable and limited to the Nigerian context.

Current applications of building automation in Nigeria should be addressed in the literature review.

The manuscript is missing detailed information about the questionnaire respondents. The employed survey represents the opinion of the respondents on the drivers and benefits of employing robotics in building industry. However, the analysis and discussion did not consider the differences in the participant groups. Therefore, analysis and findings must be categorized and linked to the respondents’ background. 

Reviewer 2 Report

·       IReferring to the abstract, to evaluate the connections between drivers and applications of robotics. How does the application fit in this context?

·       Section 2: To introduce the acronym of TAM.

·       Referring to the statement, ‘ Here robotics implementation has boost occupation safety through performing unsafe tasks in some hazardous areas for humans”

o   any supportive articles for this  statement

o   to elaborate further with  examples of robotics implementation in this context, i.e. success stories.

·       on sampling method:

o   Why stratified sampling approach selected in this study? How does this suit?

o   To explain further  the criteria selection of respondents. How do authors define the respondents as construction experts

To introduce the acronym of  MIS 

Reviewer 3 Report

As sustainability concepts should be adopted via new technologies to achieve greatest gains without compromising the objectives of the projects, this research empirically investigates the influence of identified drivers on the implementation of robots in the building sector of developing countries.

The quality of the paper and interest to readers is high. The paper uses very comprehensive and high quality references. Research background and model development as well as methodology are presented. Results are discussed and concluded. This will help the stakeholders by adopting and employing robotics by emphasizing the goal of the project concerning resources and environmental requirements, which can impact the magnitude of the project’s success.

I have no essential critical comments. To my opinion, the paper is worth to be accepted. 
